# Potential Early Markers for Breast Cancer: A Proteomic Approach Comparing Saliva and Serum Samples in a Pilot Study

**DOI:** 10.3390/ijms24044164

**Published:** 2023-02-19

**Authors:** Indu Sinha, Rachel L. Fogle, Gizem Gulfidan, Anne E. Stanley, Vonn Walter, Christopher S. Hollenbeak, Kazim Y. Arga, Raghu Sinha

**Affiliations:** 1Department of Biochemistry and Molecular Biology, Penn State University College of Medicine, Hershey, PA 17033, USA; isinha@pennstatehealth.psu.edu; 2Environmental Science and Sustainability Program, Harrisburg University of Science and Technology, Harrisburg, PA 17101, USA; rfogle@harrisburgu.edu; 3Department of Bioengineering, Marmara University, Istanbul 34854, Turkey; gizemgulfidn@gmail.com (G.G.); kazim.arga@marmara.edu.tr (K.Y.A.); 4Mass Spectrometry Core, Penn State University College of Medicine, Hershey, PA 17033, USA; aes7@psu.edu; 5Department of Public Health Sciences, Penn State University College of Medicine, Hershey, PA 17033, USA; vwalter1@pennstatehealth.psu.edu; 6Department of Health Policy and Administration, The Pennsylvania State University, University Park, State College, PA 16801, USA; csh10@psu.edu; 7Genetic and Metabolic Diseases Research and Investigation Center, Marmara University, Istanbul 34854, Turkey

**Keywords:** saliva, serum, benign, malignant, breast cancer, proteomics, protein–protein interaction network, potential biomarkers, diagnostic, prognostic

## Abstract

Breast cancer is the second leading cause of death for women in the United States, and early detection could offer patients the opportunity to receive early intervention. The current methods of diagnosis rely on mammograms and have relatively high rates of false positivity, causing anxiety in patients. We sought to identify protein markers in saliva and serum for early detection of breast cancer. A rigorous analysis was performed for individual saliva and serum samples from women without breast disease, and women diagnosed with benign or malignant breast disease, using isobaric tags for relative and absolute quantitation (iTRAQ) technique, and employing a random effects model. A total of 591 and 371 proteins were identified in saliva and serum samples from the same individuals, respectively. The differentially expressed proteins were mainly involved in exocytosis, secretion, immune response, neutrophil-mediated immunity and cytokine-mediated signaling pathway. Using a network biology approach, significantly expressed proteins in both biological fluids were evaluated for protein–protein interaction networks and further analyzed for these being potential biomarkers in breast cancer diagnosis and prognosis. Our systems approach illustrates a feasible platform for investigating the responsive proteomic profile in benign and malignant breast disease using saliva and serum from the same women.

## 1. Introduction

Breast cancer is the second leading cause of mortality for women in the United States and is estimated to result in 43,250 deaths in 2022 [1]. Early detection for breast cancer can reduce breast cancer-related mortality. Among women aged 50 years and older, reports have demonstrated a 20–40% reduction in breast cancer mortality in women who underwent mammography and clinical breast examination [2]. Among women screened at younger ages (40–49 years), mortality rates decrease by 13–23%. A detailed analysis of these data suggests that a survival rate of 96% can be achieved if women underwent mammography every three months [3]. However, the cost and risks of mammography (such as radiation exposure) with increased frequency of use are not ideal. Furthermore, despite accurate mammography diagnoses, the screening procedure may result in relatively high rates of false-positive (56%) and false-negative (22%) diagnoses in women younger than 50 years, especially in women with dense parenchymal breast tissue [4,5]. Because of these shortcomings, there is a need to develop additional diagnostic methods to further enhance the sensitivity and specificity of breast cancer detection, particularly in women with dense breast tissue, and thereby reducing the need for unnecessary biopsies. As a complementary approach to mammography, determination of biomarkers in saliva and/or serum could be a critical measurement for the early detection of breast cancer.

Saliva is considered an easily obtained clear fluid, which is indicative of an individual’s protein profile at the time of collection. Testing saliva as a diagnostic fluid meets the criteria for an inexpensive, non-invasive, reliable, and relatively simple procedure that can be repeated with a minimum discomfort to patients. In addition, providing a saliva sample may cause less anxiety in study participants than providing a blood sample [6].

The clinical utility of saliva as a diagnostic fluid is being recognized in several diseases, including cancer [7,8,9]. A meta-analysis revealed that salivary proteins represent good biomarkers for diagnosis of several cancer types including that of the breast [10,11]. Earlier studies focused on transcriptomic and proteomic signatures in saliva revealing sensitive and specific biomarkers for the detection of breast cancer using two-dimensional difference gel electrophoresis (2D-DIGE) [9]. A recent review systematically captures proteomics-based technologies for comparing dysregulated proteins in breast cancer in several body fluids including saliva and serum [12]. A variety of methods including surface enhanced laser desorption/ionization [13] and nano-liquid chromatography-quadrupole-time-of-flight technology [14] have been utilized for discovering biomarkers for breast cancer in saliva and plasma. Moreover, the isobaric tags for relative and absolute quantitation (iTRAQ) technique has been utilized for identifying salivary proteins as potential biomarkers for breast disease [15]. In a comparison study, the global-tagging iTRAQ technique was found to be more sensitive than the cysteine-specific Isotope-coded affinity tag (cICAT) method, which in turn was equally sensitive as the 2D-DIGE technique [16]. iTRAQ has an advantage over ICAT and other methods since several samples can be analyzed simultaneously, and helps reduce the time spent for mass spectrometry analysis [17]. Another advantage of iTRAQ is the possibility of identifying proteins with varying pI and molecular weights. In addition, using iTRAQ the relative and absolute quantification is possible across different sample states for a synchronous comparison of biological fluids such as saliva and serum from normal, benign and malignant breast disease cases. 

We hypothesize that protein changes occurring in breast cells and their environment will be reflected in the saliva and serum of breast cancer patients. We further hypothesize that protein changes in the benign stages will differ from those in the malignant stages of breast disease. In the present study, we compared the proteomic profile in saliva and serum samples from women without breast disease (referred to as normal in our study), with benign breast disease, and with malignant breast disease using the iTRAQ technique. Several proteins were identified in both the benign and malignant groups that could be potential biomarkers for early detection and prognosis of breast cancer in women.

## 2. Results

### 2.1. Proteins Identified in Saliva Samples

A total of 591 proteins were identified following iTRAQ analysis in the saliva samples (Appendix A). Of these, the expression of 110 proteins were statistically different (*p* < 0.05) in samples from either benign/normal (B/N), malignant/normal (M/N) or malignant/benign (M/B) comparisons (Table 1). Proteins were considered down-regulated when the pooled summary ratio was less than 1, and up-regulated when the pooled ratio was greater than 1. Additionally, 44 proteins in B/N samples (16 up-regulated, 28 down-regulated), 67 proteins in M/N samples (26 up-regulated, 41 down-regulated) and 35 proteins in M/B samples (17 up-regulated, 18 down-regulated) were observed as differentially expressed. 

It was clear that there were more down-regulated proteins in saliva samples in each comparison. Eleven proteins (ANXA1, PRELP, PRDX1, H2B2F, GSTP1, PRPC, CDC42, K2C1, PRTN3, CRNN, 6PGD) were significantly down-regulated in both B/N and M/N comparisons (*p* < 0.05). Six proteins were significantly up-regulated in both B/N and M/N (CYTS, CAH6, CATD, LG3BP, QSOX1, AMY1B) (*p* < 0.05). S10A8 was greater than 1 for benign and less than 1 for malignant diagnosis (*p* < 0.05), while 4 proteins (CYTN, AMY2B, PIGR, PERL) were high in both M/N and M/B (*p* < 0.05), 8 proteins (ANXA1, PSB3, CATG, H4, TALDO, TKT, TGM3, S10A8) were low in both M/N and M/B (*p* < 0.05) andANXA1 was down-regulated in all three comparisons. Interestingly, 10 proteins had >2 fold change in M/B (A2MG, RN150, MYO7A, PSA1, CERU, AFAM, BPIA2, SH3L1, HIS1, TTHY), 11 proteins had >2 fold change in B/N (LEG1H, DAB2P, FAM3D, CAH6, QSOX1, RAP1B, ITLN1, C251, ACSL3, S10A8, AMY1B) and 14 proteins had >2 fold change in M/N (CYTS, AMY2B, PIGR, CAH6, PLGT3, STOM, QSOX1, RETN, CYTC, AMY1B, ACTN2, AMYP, ZA2G, STAT). On the other hand, among the down-regulated proteins, 18 were <0.5 fold or less in B/N (RUSC1, SPRL1, RN150, PSA5, PRELP, ITA1, H2B2F, SPB13, CYTA, K2C6B, PSA1, GSTP1, PRPC, K2C1, PSA, PRTN3, HIS1, 6PGD), 24 proteins in M/N (ANXA1, PRELP, PSB3, NUDT5, K2C5, MMP9, CATG, PNPH, H2B2F, H4, ANXA5, ISK7, PRPC, K2C1, TGM3, ARSA, SH3L1, DEF3, S10A8, CPPED, K2C4, PRTN3, CRNN, 6PGD) and 10 proteins in M/B (PSB3, CATG, FAM3D, H2B1L, TERA, HSP76, H4, K1C10, S10AC, S10A8).

Considering the AUC values calculated from receiver operating characteristic (ROC) curve analysis for saliva proteins to distinguish between breast tumor and normal breast tissue (Table 1), 14 proteins were designated as outstanding (>90%) and 22 proteins each with excellent (80–90%) and acceptable (70–80%) ratings for their diagnostic ability [18]. 

### 2.2. Proteins Identified in Serum Samples

A total of 371 proteins were identified in the serum samples by iTRAQ analysis (Appendix A). Of these, the expressions of 56 proteins were significantly (*p* < 0.05) altered in the samples from either B/N, M/N or M/B comparisons (Table 2). In addition, 29 proteins in B/N samples (13 up-regulated, 16 down-regulated), 30 proteins in M/N samples (11 up-regulated, 19 down-regulated) and 15 proteins in M/B samples (4 up-regulated, 11 down-regulated) were observed as differentially expressed. 

Similar to saliva, a greater number of proteins were down-regulated in serum samples in each comparison. Seven proteins (APOB, TRFE, A2MG, HEP2, KAIN, TSP1, THBG) were significantly down-regulated in both B/N and M/N comparisons (*p* < 0.05) and 4 proteins (PRDX2, A1BG, FIBA, APOH) were significantly up-regulated in both B/N and M/N (*p* < 0.05). HBB was up-regulated while 4 proteins (TRFE, APOA1, TSP1, APOA2) were down-regulated in both M/N and M/B comparisons. TSP1 was down-regulated in all the three comparisons. In addition, 4 proteins (HBB, VINC, CD5L, PCD20) showed more than 1.5-fold change in M/B, 8 proteins (DYST, VWF, CO6, PRDX2, A1BG, LUM, CE290, APOH) were changed by >1.5 fold in B/N, and 7 proteins (HBB, PRDX2, A1AG1, A1BG, BLVRB, FIBA, APOH) were up-regulated by >1.5 fold in M/N. On the other hand, 4 proteins (TRFE, CATD, COL11, DYHC1) were down-regulated by 0.5 fold or lower in B/N, 6 proteins (TRFE, SOX, TSP1, MED30, A1AT, SMC3) were down-regulated in M/N and 3 proteins (APOA1, GPKOW, ERBIN) were down-regulated in M/B. 

AUC values for serum samples indicated that 9 proteins demonstrated outstanding (>90%), 8 proteins showed excellent (80–90%) and 7 proteins showed acceptable (70–80%) diagnostic performance (Table 2). 

### 2.3. Enrichment Analysis of Proteins in Saliva Samples

GO enrichment analysis showed that in all three comparisons (B/N, M/N and M/B), most salivary proteins were involved in exocytosis, secretion, immune response, neutrophil mediated immunity and cytokine-mediated signaling pathway, but the number of proteins associated with these processes varied between groups (Figure 1A). Most proteins were localized in the extracellular exosome, extracellular space, secretory granule lumen, secretory vesicle or cytoplasmic vesicles, and again the number of proteins varied among the groups. In terms of molecular functions, the proteins were annotated as enzyme inhibitor activity, calcium ion binding, endopeptidase regulator activity and peptidase activity (Figure 1A, Appendix A).

KEGG pathway analysis identified a total of 25, 9 and 16 pathways (*p* < 0.05) and Reactome pathway analysis identified 44, 36 and 25 pathways (*p* < 0.05) for the B/N, M/N and M/B groups of saliva samples, respectively. The overall comparison among the groups can be found in Appendix A. The top 10 enriched Reactome pathways related to each of the group samples are shown for B/N (Figure 1B), M/N (Figure 1C) and M/B (Figure 1D) related to the significant proteins in each group. The saliva proteins identified from iTRAQ analysis of B/N, M/N and M/B groups were mainly involved in the neutrophil degranulation and innate immune response based on Reactome pathway analysis (*p* < 0.05).

### 2.4. Enrichment Analysis of Proteins in Serum Samples

The serum proteins were mostly involved in the regulation of biological processes, were located primarily in organelles or extracellular region, and mostly displayed binding, catalytic or structural molecular activities (Figure 2A, Appendix A).

KEGG pathway analysis identified a total of 10, 11 and 13 pathways (*p* < 0.05) and Reactome pathway analysis identified 59, 45 and 43 pathways (*p* < 0.05) for B/N, M/N and M/B group of samples, respectively (Figure 2B–D, Appendix A). The serum proteins identified from iTRAQ analysis of B/N group were mainly involved in platelet degranulation (*p* = 1.64 × 10^−12^), response to elevated platelet cytosolic Ca^2+^ (*p* = 2.31 × 10^−12^), platelet activation, signaling and aggregation (*p* = 8.61 × 10^−10^). While M/N group serum proteins were engaged in chylomicron assembly (*p* = 2.90 × 10^−9^), chylomicron remodeling (*p* = 2.90 × 10^−9^), and retinoid metabolism and transport (*p* = 5.34 × 10^−8^). Further, in the serum samples from M/B group the proteins were also involved in similar proteins as B/N group with lesser *p* values; platelet degranulation (*p* = 1.24 × 10^−9^), response to elevated platelet cytosolic Ca^2+^ (*p* = 1.56 × 10^−9^), platelet activation, signaling and aggregation (*p* = 8.13 × 10^−8^).

### 2.5. Protein-Protein Interaction (PPI) Networks for Proteins in Saliva

The network of B/N consisted of 798 interactions among 28 significant saliva proteins and 602 of their first interacting neighbors. Among the major hub proteins in the B/N group, CDC42, HSP7C, PSA1 and PSA5 were down-regulated and had multiple interacting partners, whereas S10A8, CATD, FINC and LG3BP were up-regulated with a moderate number of interactions (Figure 3). The network of M/N consisted of 620 interactions among 44 significant proteins and 521 of their first interacting neighbors. The major hubs in the M/N group consisted of CDC42, H2AX, PSB3 and PDIA1 which were down-regulated and had several to moderate interacting partners while LGS3BP, STOM, ACTN2 and VPS41 were up-regulated with fewer interacting partners (Figure 3). In addition, the network of M/B consisted of 522 interactions among 19 significant proteins and 407 of their first interacting neighbors. Among the major hub proteins in the M/B group, TERA, FINC, HSP7C, PSB3, S10AB and ANXA1 were down-regulated and had several to moderate interacting partners, whereas HS71A, PSA1, A2MG and TTHY were up-regulated with a moderate number of interactions (Figure 3). All the PPIs in each of the groups in saliva are listed in Appendix A.

### 2.6. Protein–Protein Interaction Networks for Proteins in Serum

Overall, there were fewer interactions in serum among the smaller number of significant proteins and far fewer interacting partners compared to the respective PPI networks among saliva proteins. In particular, the network of B/N consisted of 161 interactions among 20 significant proteins and 151 of their first interacting neighbors. Among the major hub proteins in the B/N group, DYHC1, APOB, CATD, TSP1 and A2MG were down-regulated and had moderate interacting partners and were down-regulated, whereas CE290, PRDX2, CADH5 and APOC1 were up-regulated with a moderate number of interactions (Figure 3). The network of M/N consisted of 175 interactions among 20 significant proteins and 165 of their first interacting neighbors. The major hubs in the M/N group consisted of MED30, SMC3, APOB and APOA1, which were down-regulated and had several to moderate number of interacting partners, whereas PRDX2, HBB, FIBA and FETUA were up-regulated with fewer interacting partners (Figure 3). Additionally, the network of M/B consisted of 66 interactions among 11 significant proteins and 61 of their first interacting neighbors. Among the major hub proteins in the M/B group, APOA1, APOA2, GPKOW and TSP1 were down-regulated and had moderate interacting partners whereas VINC and HBB were up-regulated with a moderate number of interactions (Figure 3). All the PPIs in each of the groups for the serum samples are listed in Appendix A.

### 2.7. Protein Ratios across Serum and Saliva in B/N and M/N Groups

Following the iTRAQ analysis, proteins commonly identified in saliva and serum samples of the B/N and M/N groups were fitted without interaction by two-way ANOVA models. As a result, we identified 17 proteins that were significantly (*p* < 0.05) different among serum and saliva (Appendix A). A subset of these proteins that were detected in 6–8 saliva and serum samples are shown in Figure 4. These included alpha-1B-glycoprotein precursor (A1BG), fibrinogen alpha chain isoform alpha-E preproprotein (FIBA), alpha-1-antichymotrypsin precursor (AACT), extracellular matrix protein 1 isoform 3 precursor (ECM1), peroxiredoxin-2 (PRDX2), 78 kDa glucose regulated protein precursor (ERP78) and galactin-3-binding protein precursor (LG3BP). PRDX2, A1BG, ECM1, ERP78 and FIBA showed lower ratios in saliva samples when compared to serum samples while LG3BP and AACT ratios were higher in saliva in contrast to serum samples of the same subjects. Upon further comparison between B/N and M/N in the saliva and serum samples, TSP1 was found to be significantly different in serum (*p* < 0.05). All the above proteins were presently measured as a ratio following iTRAQ analysis and need to be validated using actual quantitation by either Western blot analysis or ELISA in the future.

### 2.8. Prognostic Performance Analysis

When the association of the expression levels of genes encoding significant proteins with prognostic outcome was investigated through survival analyses, all protein sets in B/N and M/N saliva and serum samples (Figure 5), except in the M/B group serum data, indicated high impact on overall patient survival (*p* < 0.05) in breast cancer. According to the parameters of HR and *p*-values, the prognostic performance of the protein sets in the saliva data was observed to be more significant than the protein sets in the serum data for all the groups. In addition, the comparisons of the B/N and M/N group samples had better prognostic performance than the M/B group samples in both the saliva and serum data. The prognostic performance of each gene encoding significant protein based on high-risk vs. low-risk of the dataset for invasive breast carcinoma (BRCA) obtained from The Cancer Genome Atlas (TCGA) were used to draw the Kaplan–Meier (KM) plots and are presented as box plots for the significant proteins in B/N, M/B and M/N groups in saliva as well as in serum (Appendix A).

## 3. Discussion

When comparing the proteomic profile of saliva and serum samples from the same women with a diagnosis of benign or malignant state of the breast disease relative to those of women with no breast disease, we have identified proteins that differed in expression levels. Further, analyzing the significant protein changes suggested involvement of several biological pathways and functionalities. We constructed potential protein–protein interaction networks among hub proteins detected in serum and saliva samples and their interacting partners to identify potential biomolecular markers to be explored for diagnosis or prognosis. Since mammography can lead to false positives and anxiety in subjects, utilizing more than one biomarker from our analysis would greatly improve early diagnosis of breast cancer using non-invasive testing in saliva and/or serum.

Interestingly, several proteins in our saliva and serum proteomic analysis qualified for outstanding and excellent diagnostic power based on the AUC values (Table 1 and Table 2). However, ROC curve analysis was based on RNA-Seq data from breast tumor tissues compared to normal tissues from the TCGA database; therefore, it is worthwhile to investigate which of these secretory proteins succeed as biomarkers for early breast cancer diagnosis using a larger cohort.

Several circulating proteins have been identified in the plasma and serum of patients with breast cancer [19] but we still lack highly sensitive and specific biomarkers. Below, we discuss some of the pertinent proteins that were significantly altered among the different groups (B/N, M/M and M/B) in our analysis of saliva and/or serum and their relevance for a potential biomarkers for breast cancer. 

**Saliva:** Fibronectins bind cell surfaces and various compounds including collagen, fibrin, heparin, DNA and actin. In our analysis, fibronectin isoform 11 preproprotein (FINC), was up-regulated 1.97 fold in B/N (*p* < 0.05) and did not change in M/N, whereas it was down-regulated at 0.56 fold in the M/B group (*p* < 0.05). It has been reported that a liquid biopsy detecting FINC on circulating extracellular vesicles could be a promising method to detect early breast cancer [20]. Indeed, FINC was one of the hub proteins that had 15 interacting partners and has an AUC of 93.05% with an outstanding diagnostic power.

The SPARK-like isoform 1 protein 1 precursor (SPRL1) is an extracellular matrix glycoprotein that has been implicated in the pathogenesis of several disorders, including cancer. In our analysis, SPRL1 was down-regulated at 0.15 fold (*p* < 0.05) in B/N and 0.44 fold in M/N (*p* > 0.05). Previously, a significantly reduced expression SPRL1 was observed in human breast cancer tissues compared to that in normal breast epithelial tissues, at both mRNA and protein levels. In addition, the down-regulation of SPRL1 was significantly correlated with lymphatic metastasis [21]. SPRL1 was found to have an outstanding diagnostic power with an AUC of 96.5%.

Histone H2AX (H2AX) is a type of histone protein from the H2A family encoded by the H2AFX gene. An important phosphorylated form is γH2AX (S139), which forms when double-strand breaks appear. In our analysis, H2AX was marginally up-regulated at 1.16 fold in B/N (*p* > 0.05) but down-regulated significantly in M/N at 0.5 fold (*p* < 0.05) and at 0.39 fold in M/B group (*p* > 0.05). Evaluating the formation of γH2AX in breast tumor tissue could potentially be a sensitive means of early breast cancer detection as these levels may reflect endogenous genomic instability in breast cancerous tissues [22]. Additionally, the detection of γH2AX could benefit early cancer screening, with breast cancer included [23]. Even though in our analysis we found H2AX to be down-regulated in M/N group, it is important to note that we detected H2AX in saliva and this could be conveniently used for monitoring breast disease. H2AX was one of the hub proteins that had 102 interacting partners and had an AUC of 93.7% with an outstanding diagnostic power.

Cystatin-SN precursor (CYTN) belongs to the type 2 cystatin superfamily, which restricts the proteolytic activities of cysteine proteases. In our analysis, CYTN was marginally up-regulated at 1.94- and 1.96-fold in M/N and M/B groups, respectively, while only a moderate change of 1.09 was noted in benign samples (*p* > 0.05). CYTN promotes cell proliferation, clone formation and metastasis in breast cancer cells and has been proposed to be a potential prognostic biomarker and therapeutic target for breast cancer [24]. CYTN was found to have an outstanding diagnostic power with AUC of 93.1%.

**Serum:** Hemoglobin subunit beta (HBB) is a member of the globin family, a structurally conserved group of proteins often containing the heme group, which have the ability to reversibly bind O2 and other gaseous ligands in erythrocytes [25]. In our analysis, HBB was up-regulated 1.97- and 2.04-fold in M/N and M/B groups, respectively (*p* < 0.05), but moderately down-regulated in B/N group. This protein has been implicated as a potential biomarker of breast cancer progression [26]. It was one of the hub proteins that had 5 interacting partners and had an AUC of 93.7% with outstanding diagnostic power. 

Retinol-binding protein 4 (RET4) is a recently identified adipokine that is elevated in patients with obesity or type 2 diabetes [27]. The iTRAQ analysis revealed that RET4 was up-regulated 1.48 fold in M/N group (*p* < 0.05) and may be detectable earlier as suggested from our study (1.40 fold increase in B/N, *p* > 0.05). In a case control study, higher serum RET4 levels were associated with the risk of breast cancer [28]. It was one of the hub proteins with just 1 interacting partner (TTHY) and had an AUC of 93.5% with outstanding diagnostic power.

Cadherin-5 isoform X1 (CADH5) is a member of the cadherin family which are calcium-dependent cell adhesion proteins. Previously, using a glycoproteomic approach CADH5 emerged as a novel biomarker for metastatic breast cancer [29]. The iTRAQ analysis revealed that CADH5 was up-regulated 1.18 fold in M/N group (*p* > 0.05) and was most likely detected in the benign stage of breast cancer as suggested from our study (1.50 fold increase in B/N, *p* < 0.05). It was one of the hub proteins with 8 interacting partners and has an AUC of 91.6% with outstanding diagnostic power.

Von Willebrand factor preproprotein (VWF) is a large multimeric plasma glycoprotein that plays important roles in normal hemostasis. VWF can also impact cancer cell metastasis [30] and more recently it has been shown by the same group that breast cancer cells mediate endothelial cell activation and promote VWF release [31]. However, in our analysis VWF was elevated 1.57 fold in serum samples of benign breast cancer diagnosis (*p* < 0.05), so this may be a potential marker that may provide damage to endothelial cells early in the disease. It was one of the hub proteins with 4 interacting partners and had an AUC of 91.6%.

Alpha-2-macroglubulin isoform X1 (A2MG) is a protease inhibitor and cytokine transporter covering a wide range of proteases, including trypsin, thrombin and collagenase. Even though it has a high AUC value for diagnosis (92.4%), it was modestly down-regulated in both benign and malignant samples (*p* < 0.05). Others have reported it to be lower [32] or higher [14] in breast cancer.

Peroxiredoxin-2 (PRDX2), and peroxiredoxins in general, catalyze the reduction reaction of peroxide and maintain the balance of intracellular H_2_O_2_ levels. In our analysis, PRDX2 was up-regulated 1.89- to 2.16-fold in B/N and M/N groups, respectively (*p* < 0.05), but exhibited no change in M/B group. High mRNA expression of PRDX1/2/4/5/6 was significantly associated with shorter relapse-free survival in breast cancer patients [33]. It was one of the hub proteins that had 16 interacting partners and had an AUC of 80.2% with excellent diagnostics power.

Among the proteins commonly identified across serum and saliva, PRDX2, LG3BP and TSP1 are promising for further investigation to distinguish the benign from the malignant stage of breast cancer in a larger cohort. Moreover, some of the proteins identified in the present study have been associated with Hallmarks of Cancer specific proteins in breast cancer [34], including FINC, proteasome subunit alpha type-1 isoform 2 (PSA1), proteasome subunit alpha type-5 isoform 1 (PSA5), proteasome subunit beta type-3 (PSB3), phosphoglycerate kinase 1 (PGK1), heat shock cognate 71 kDa protein isoform 1 (HSP7C) and glutathione S-transferase *p* (GSTP1) which may provide insights into the early detection of breast disease.

We have further identified several proteins in saliva, including AMY1B, AMY2B, BPIB2, CPPED, DEF3, H2A2A, H2BC18, ISK7, LEG1H, PNP, PRELP, SPB13, STAT, QSOX1, RNF150 and VPS41, and in serum, namely, CEP290, CO8B, CO6, CPN2, GPKOW, HEP2 and PIPOX which have not been reported in the literature to previously be associated with breast cancer. These are suggestive potential candidate biomarkers for the early detection of breast cancer.

## 4. Materials and Methods

### 4.1. Study Subjects

Subjects were recruited at the Hershey Medical Center Breast Cancer Center upon their routine visit for a mammogram. Sixty healthy adult women with no breast disease, 13 adult women with a diagnosis of benign breast disease and 15 adult women with a diagnosis of malignant breast disease were enrolled in the study. All participants provided written informed consent, following the protocol approved by the Pennsylvania State University Institutional Review Board (STUDY00005159). Subjects were recruited based on the following inclusion criteria: English-speaking female volunteers, 25–85 years of age, who had undergone mammogram examination and were currently non-smokers. Exclusion criteria included any evidence of cancer other than the breast and undergoing treatment for breast cancer prior to saliva and blood sample collection. When there was any abnormality detected on the mammogram, subjects were advised to undergo a biopsy. The diagnosis on the breast biopsy tissues following the surgical pathology reporting on Hematoxylin and Eosin-stained sections were provided by Board Certified Pathologists in the Department of Pathology, at the Penn State College of Medicine. Table 3 provides the characteristics of the subjects used for iTRAQ analysis.

### 4.2. Collection and Storage of Biological Samples

Saliva and blood samples were collected in the fasting state. Saliva samples were centrifuged at 10,000 rpm for 15 min at 4 °C and the clear supernatants were aliquoted in 1 mL screw capped vials. For serum, clotted blood was separated and centrifuged at 1300 rpm for 15 min at 4 °C. The clear serum was aliquoted in 1 mL screw cap vials. All biological samples were stored at −80 °C until analyzed. 

### 4.3. Sample Processing and Labeling Procedure for iTRAQ Analysis 

Eight saliva and serum samples from the participants in each group were processed for iTRAQ analysis as described earlier [35,36]. The serum samples but not the saliva samples were depleted of the 6 most abundant proteins including albumin, IgG, IgA, transferrin, haptoglobin and antitrypsin using a Multiple Affinity Removal System LC Column (Agilent Technologies, Santa Clara, CA). Briefly, equal amounts of protein (100 μg) from each sample were digested with trypsin and subsequently labeled with one of 8 unique isobaric tags using the iTRAQ^®^ Reagent-8Plex Multiplex kit (AB SCIEX, Framingham, MA). Quantitative fragments, ranging from 113 to 121 Daltons, following MS/MS fragmentation shows proportionally how much of each peptide peak came from each of the individually labeled samples. The Penn State College of Medicine’s Proteomic Core Facility received the tagged samples which were subsequently resolved by two-dimensional liquid chromatography prior to triple time-of-flight (TOF) mass spectrometry. Peptide identification, protein grouping and subsequent protein quantitation were done using the Paragon algorithm as implemented in Protein Pilot 5.0 software (ProteinPilot 5.0, which contains the Paragon Algorithm 5.0.0.0, build 4632 from ABI/MDS- SCIEX), searching the NCBI human database plus a list of 389 common contaminants (see Appendix B for details). The datasets presented in this software are ratios of samples with defined diagnoses (e.g., B/N, M/N or M/B). Ratios significantly greater than 1 in a B/N ratio indicates a differential increase in protein in benign compared to normal (similarly for M/N and M/B), and ratios significantly less than 1 in a B/N ratio indicates a differential decrease in benign compared to normal (similarly for M/N and M/B).

### 4.4. Gene Set Over-Representation Analysis

Functional annotations associated with the significant proteins determine as a result B/N, M/N and M/B comparisons in the saliva and serum data were identified in terms of biological processes, signaling and metabolic pathways by over-representation analyses using the Consensus PathDB [37]. As the sources for pathway databases, Kyoto Encyclopedia of Genes and Genomes (KEGG) [38] and Reactome [39] were used. While the annotation of the biological process, cellular components and molecular function were determined using Gene Ontology (GO) [40] annotations. The significance of over-representations was evaluated by adjusted-*p*-values via Fisher’s Exact Test, followed by Benjamini-Hochberg correction. Functional enrichment results with an adjusted *p*-value < 0.05 were considered statistically significant.

### 4.5. Construction of Protein–Protein Interaction Network

Using physical protein–protein interaction (PPI) data consisting of 68,948 interactions among 10,835 proteins which were experimentally detected in human and stored in the BioGRID database (version 4.4.210) [41], PPI networks were constructed around the significant proteins found in three comparisons (B/N, M/N, M/B) in saliva and serum data by enriching them with their first-neighbor interactions. The visualization of the PPI networks was performed via Cytoscape (v.3.7.0) software [42].

### 4.6. Prognostic Performance Analysis

The prognostic performance of the significant proteins in all three comparisons (B/N, M/N, M/B) for saliva and serum data were assessed with survival analyses according to the established pipeline [43,44] using RNA-Sequencing (RNA-Seq) data from 1102 patients suffering from invasive breast carcinoma obtained from TCGA database. Each individual was classified into low- and high-risk groups according to their risk score, the prognostic index (PI), according to the linear component of the Cox model with the equation
PI = β_1×1_ + β_2×2_ +… + β_p×p_
(1)
where x_i_ is the expression value of each gene and β_i_ is the coefficient obtained from the Cox fitting. Survival analyses were performed using the survival package (v.3.3.1) [45] in R (v.4.0.4). KM survival plots provided visualizations for the survival time statistics calculated by log-rank test, and the log-rank *p*-value < 0.05 was considered as the cut-off to describe the statistical significance of survival in each group. In addition, the HR was calculated to quantify the relative hazard of each KM plot.

### 4.7. ROC Curve Analysis

ROC curve analysis was performed for each significant protein in each of the three comparisons (B/N, M/N, M/B) for saliva and serum data using RNA-Seq data of the BRCA dataset including 1102 tumor and 113 normal samples in order to assess the diagnostic capability of protein markers to discriminate between individuals. The AUC values for each ROC curve were measured to determine how well it can discriminate between two diagnostic groups (tumor and normal). In general, a ROC = 0.5 suggests no discrimination, 0.7 ≤ ROC < 0.8 suggests acceptable discrimination, 0.8 ≤ ROC < 0.9 suggests excellent discrimination and ROC ≥ 0.9 is considered outstanding discrimination [18]. ROC analyses were performed via pROC package [46] in R. 

### 4.8. Statistical Analyses 

To combine protein ratios across separate iTRAQ runs and to determine whether protein ratios differed significantly between the three comparisons (B/N, M/N and M/B), the ratios were modeled using a random effects model described earlier [35,47]. 

Briefly, this procedure used a weighted average of individual ratios across multiple iTRAQ runs to estimate an overall ratio. The weights were proportional to the inverse of the variance for each individual run. The overall protein ratio was deemed statistically significant at the 5% level if the 95% confidence interval did not contain 1. Proteins identified in multiple iTRAQ experiments with ratios that were statistically significant after combining across runs were considered proteins of interest. Algorithms for combining proteins were programmed using the rmeta package in R.

Additionally, for comparing protein ratios across serum and saliva in B/N and M/N samples, two-way ANOVA models without interaction were fitted using individual iTRAQ protein log2(ratios) obtained from saliva and serum samples for either benign (B) or malignant (M) compared to normal (N) samples. The *car* R package [48] was utilized to perform the marginal test comparing saliva and serum mean protein log2(ratios) while controlling for diagnosis (benign or malignant). Two sample t-tests were performed to compare iTRAQ protein log2(ratios) for select proteins in M/N vs. B/N groups after restricting to samples from either saliva or serum. Statistical significance was assessed at the *α* = 0.05 level. Because of the exploratory nature of this study, no adjustment for multiple testing was applied. R 4.0.5 [49] was used to create box plots and perform all statistical analyses.

## Figures and Tables

**Figure 1 ijms-24-04164-f001:**
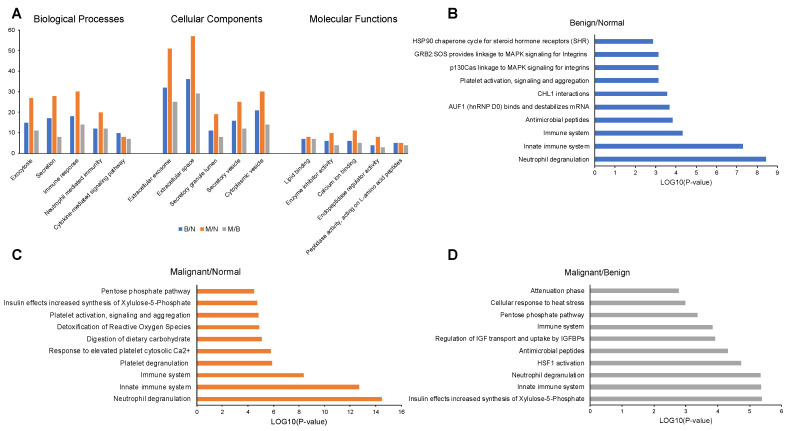
GO classification and enrichment analysis of significantly expressed proteins in saliva samples of B/N, M/N and M/B groups. (**A**) GO classification in biological processes, cellular components and molecular functions. Top 10 enriched Reactome pathways for (**B**) B/N, (**C**) M/N and (**D**) M/B groups.

**Figure 2 ijms-24-04164-f002:**
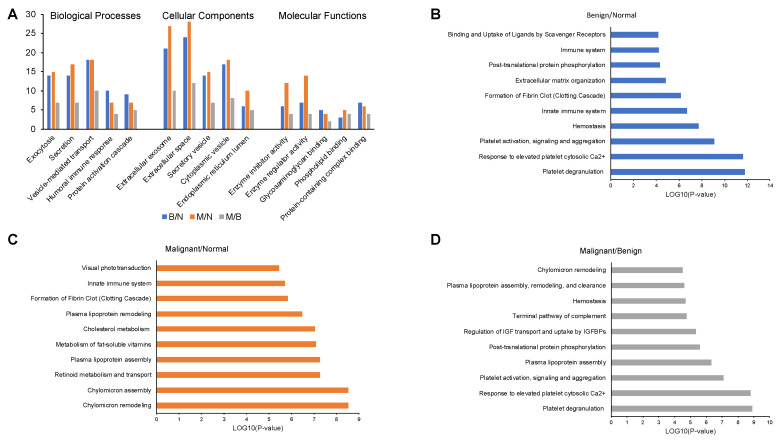
GO classification and enrichment analysis of significantly expressed proteins in serum samples of B/N, M/N and M/B groups. (**A**) GO classification in biological processes, cellular components and molecular functions. Top 10 enriched Reactome pathways for (**B**) B/N, (**C**) M/N and (**D**) M/B groups.

**Figure 3 ijms-24-04164-f003:**
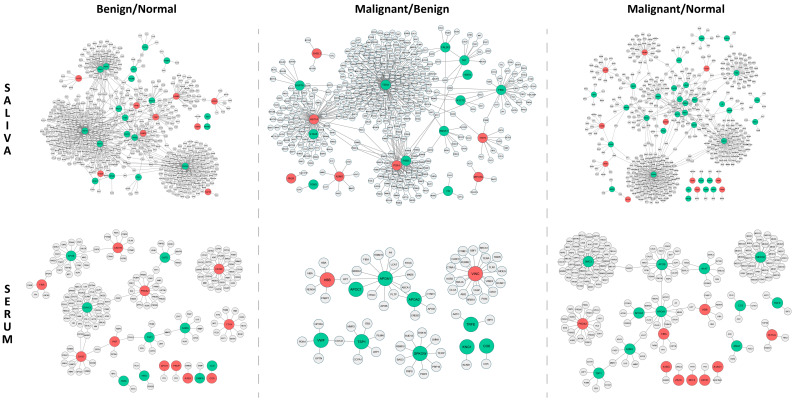
PPI networks for significant proteins in saliva and serum in B/N, M/B and M/N groups. Proteins colored in red are up-regulated and green colored proteins are down-regulated in expression.

**Figure 4 ijms-24-04164-f004:**
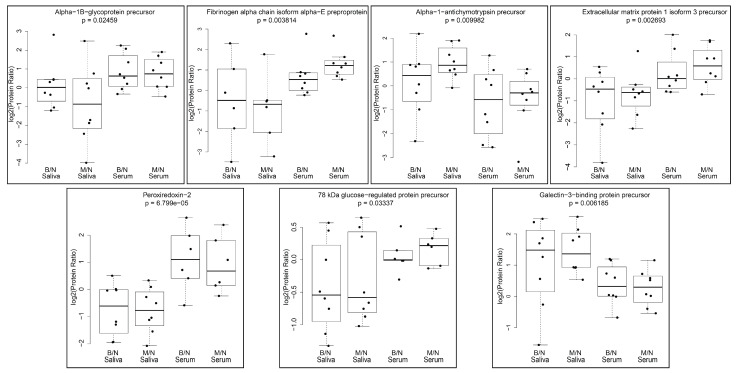
Protein ratios in B/N and M/N groups compared for common proteins identified in iTRAQ analysis. Box plots for significant proteins in 6–8 saliva and serum samples are displayed.

**Figure 5 ijms-24-04164-f005:**
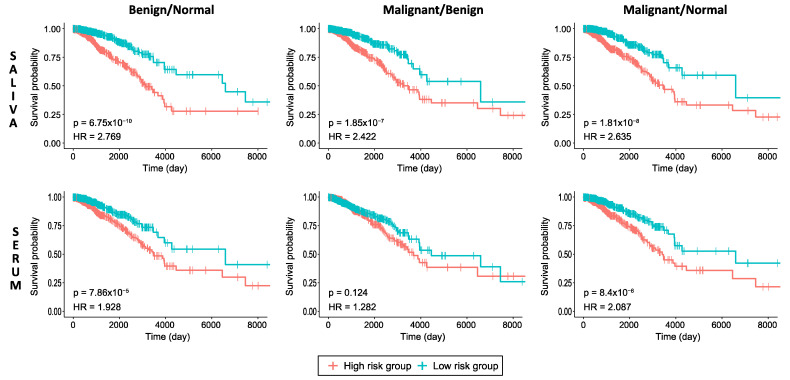
KM plots estimating patients’ survival based on significant proteins in B/N, M/B and M/N groups for breast cancer indicating *p*-value of the log-rank test and HR for each curve. The censoring samples are shown as “+” marks. Horizontal axis represents time to event.

**Table 1 ijms-24-04164-t001:** Significant saliva proteins identified in normal, benign and malignant samples.

Accession #	UniProt ID	Gene ID	Protein Name	B/N	M/N	M/B	Runs	Peptides (95%)	AUC (%)
189083772	GELS	GSN	Gelsolin isoform b	0.71	**0.62**	0.98	4	13	97.37
4502101	ANXA1	ANXA1	Annexin A1	**0.56**	**0.28**	**0.54**	4	12	97.23
157412247	RUSC1	RUSC1	RUN and SH3 domain-containing protein 1 isoform b	**0.23**	0.69	1.61	2	1	97.09
190341024	SPRL1	SPARCL1	SPARC-like protein 1 isoform 1 precursor	**0.15**	0.44	2.02	2	3	96.53
58331204	RN150	RNF150	RING finger protein 150 precursor	**0.30**	2.30	**7.76**	2	1	95.93
4504253	H2AX	H2AFX	Histone H2AX	1.16	**0.50**	0.39	2	5	93.68
19882251	CYTN	CST1	Cystatin-SN precursor	1.09	**1.94**	**1.96**	8	140	93.07
807066348	FINC	FN1	Fibronectin isoform 11 preproprotein	**1.97**	0.99	**0.56**	6	14	93.05
23110942	PSA5	PSMA5	Proteasome subunit alpha type-5 isoform 1	**0.27**	0.56	2.05	2	3	92.87
578822814	A2MG	A2M	Alpha-2-macroglobulin isoform X1	0.99	3.21	**4.52**	2	54	92.39
66932947	A2MG	A2M	Alpha-2-macroglobulin precursor	0.83	**0.57**	0.84	6	54	92.39
58219024	LEG1H	LEG1	Protein LEG1 homolog precursor	**2.51**	1.71	0.63	8	53	91.10
4503105	CYTT	CST2	Cystatin-SA precursor	1.35	**1.73**	1.82	8	43	90.65
94967023	DAB2P	DAB2IP	Disabled homolog 2-interacting protein isoform 1	**3.36**	1.64	0.48	2	1	90.03
4506041	PRELP	PRELP	Prolargin precursor	**0.13**	**0.33**	2.52	2	1	89.87
4503109	CYTS	CST4	Cystatin-S precursor	**1.71**	**2.73**	1.58	8	105	89.74
4826898	PROF1	PFN1	Profilin-1	0.84	**0.69**	0.75	8	9	88.24
20070125	PDIA1	P4HB	Protein disulfide-isomerase precursor	0.75	**0.67**	0.87	8	21	87.90
10280622	AMY2B	AMY2B	Alpha-amylase 2B precursor	1.86	**3.27**	**1.63**	8	765	87.89
22538465	PSB3	PSMB3	Proteasome subunit beta type-3	0.95	**0.20**	**0.31**	4	1	87.87
32455266	PRDX1	PRDX1	Peroxiredoxin-1	**0.68**	**0.65**	0.88	8	8	87.56
31657142	ITA	ITGA1	Integrin alpha-1 precursor	**0.11**	0.45	4.25	2	1	86.93
4505763	PGK	PGK1	Phosphoglycerate kinase 1	0.72	**0.71**	0.89	8	19	86.28
767959648	NUDT5	NUDT5	ADP-sugar pyrophosphatase isoform X2	0.43	**0.36**	0.82	2	1	83.99
119395754	K2C5	KRT5	Keratin, type II cytoskeletal 5	0.97	**0.41**	0.37	4	16	83.71
74272287	MMP9	MMP9	Matrix metalloproteinase-9 preproprotein	0.95	**0.48**	0.67	8	11	83.20
767909532	PIGR	PIGR	Polymeric immunoglobulin receptor isoform X1	1.82	**3.07**	**1.47**	8	125	83.20
767979880	CATG	CTSG	Cathepsin G isoform X1	0.72	**0.41**	**0.49**	8	6	83.01
767922296	FAM3D	FAM3D	Protein FAM3D isoform X2	**4.16**	1.41	**0.29**	4	2	82.75
395132469	CAH6	CA6	Carbonic anhydrase 6 isoform 2 precursor	**3.00**	**2.94**	1.15	8	36	82.20
4503143	CATD	CTSD	Cathepsin D preproprotein	**1.82**	**1.77**	1.12	8	9	82.09
157168362	PNPH	PNP	Purine nucleoside phosphorylase	0.54	**0.33**	0.53	4	8	81.62
4504259	H2B1L	HIST1H2BL	Histone H2B type 1-L	1.02	0.39	**0.36**	2	7	81.51
4758950	PPIB	PPIB	Peptidyl-prolyl cis-trans isomerase B precursor	**1.66**	1.24	0.82	8	7	80.99
6005942	TERA	VCP	Transitional endoplasmic reticulum ATPase	1.61	0.59	**0.36**	6	9	80.69
32189392	PRDX2	PRDX2	Peroxiredoxin-2	0.65	**0.59**	1.00	8	7	80.24
34419635	HSP76	HSPA6	Heat shock 70 kDa protein 6	1.39	0.42	**0.30**	2	6	79.02
66912162	H2B2F	H2BC18	Histone H2B type 2-F isoform a	**0.36**	**0.33**	0.63	6	7	78.99
8393956	SPB13	SERPINB13	Serpin B13 isoform 2	**0.45**	0.58	1.51	6	8	78.79
4885165	CYTA	CSTA	Cystatin-A	**0.35**	0.50	1.20	8	13	78.59
767969637	PLGT3	POGLUT3	Protein O-glucosyltransferase 3 isoform X1	2.44	**3.34**	1.36	2	1	76.93
4557581	FABP5	FABP5	Fatty acid-binding protein, epidermal	**0.58**	0.71	1.07	8	18	76.03
4504309	H4	HIST1H4C	Histone H4	0.96	**0.43**	**0.34**	8	6	75.67
4502107	ANXA5	ANXA5	Annexin A5	0.68	**0.27**	0.40	2	1	74.55
11496281	KLK13	KLK13	Kallikrein-13 precursor	1.41	1.17	**0.45**	6	4	74.51
530412176	K1C10	KRT10	Keratin, type I cytoskeletal 10 isoform X1	0.91	0.35	**0.34**	8	17	73.38
38016911	STOM	STOM	Erythrocyte band 7 integral membrane protein isoform a (stomatin isoform a)	3.21	**4.06**	1.26	2	1	72.54
5729877	HSP7C	HSPA8	Heat shock cognate 71 kDa protein isoform 1	**0.71**	0.66	0.86	8	21	72.46
119703753	K2C6B	KRT6B	Keratin, type II cytoskeletal 6B	**0.26**	0.77	2.90	4	24	71.90
767968230	MYO7A	MYO7A	Unconventional myosin-VIIa isoform X11	0.5	1.57	**3.08**	2	2	71.69
5031857	LDHA	LDHA	L-lactate dehydrogenase A chain isoform 1	0.92	**0.67**	0.84	8	20	71.52
53793688	H32	H3C15	Histone H3.2	1.19	0.7	**0.57**	6	1	71.31
4506179	PSA1	PSMA1	Proteasome subunit alpha type-1 isoform 2	**0.39**	0.87	**2.22**	4	1	71.04
21071008	TCO1	TCN1	Transcobalamin-1 precursor	1.09	**1.80**	1.30	8	16	70.72
4504183	GSTP1	GSTP1	Glutathione S-transferase P	**0.48**	**0.59**	1.09	8	27	70.68
5032059	S10AC	S100A12	Protein S100-A12	1.30	0.52	**0.39**	4	2	70.61
4504251	H2A2A	H2AC18	Histone H2A type 2-A	0.75	**0.54**	0.77	6	5	70.34
5902134	COR1A	CORO1A	Coronin-1A	1.01	**0.51**	0.57	8	8	70.30
5803187	TALDO	TALDO1	Transaldolase	0.90	**0.53**	**0.61**	8	14	69.63
194248072	HS71A	HSPA1A	Heat shock 70 kDa protein 1A	**0.61**	0.81	**1.25**	8	34	69.37
115387104	AL9A1	ALDH9A1	4-trimethylaminobutyraldehyde dehydrogenase	**0.67**	0.85	1.23	8	5	68.33
5031863	LG3BP	LGALS3BP	Galectin-3-binding protein precursor	**1.88**	**1.94**	1.11	8	12	68.07
4557485	CERU	CP	Ceruloplasmin precursor	0.53	0.99	**2.22**	2	16	67.22
13325075	QSOX1	QSOX1	Sulfhydryl oxidase 1 isoform a precursor	**8.00**	**7.74**	1.14	2	4	66.46
14211875	ISK7	SPINK7	Serine protease inhibitor Kazal-type 7 precursor	1.00	**0.36**	0.53	4	1	66.16
604723334	PRPC	PRH1	Salivary acidic proline-rich phosphoprotein 1/2 isoform b	**0.12**	**0.34**	0.32	2	58	66.13
7661678	RAP1B	RAP1B	Ras-related protein Rap-1b isoform 1 precursor	**4.77**	1.73	0.88	2	1	65.41
4757952	CDC42	CDC42	Cell division control protein 42 homolog isoform 1 precursor	**0.51**	**0.51**	0.98	6	2	65.13
4501987	AFAM	AFM	Afamin precursor	0.06	0.7	**4.57**	2	2	63.81
119395750	K2C1	KRT1	Keratin, type II cytoskeletal 1	**0.30**	**0.28**	0.74	2	15	63.79
15055535	BPIB2	BPIFB2	BPI fold-containing family B member 2 precursor	1.73	**1.94**	1.21	8	34	62.36
9966777	RETN	RETN	Resistin precursor	1.38	**2.87**	1.91	4	2	62.34
31542986	ITLN1	ITLN1	Intelectin-1 precursor	**6.64**	2.70	0.40	2	2	62.13
5454052	1433S	SFN	14-3-3 protein sigma	**0.67**	0.81	1.30	8	26	60.75
530417837	BPIA2	BPIFA2	BPI fold-containing family A member 2 isoform X1	0.58	1.34	**2.23**	8	35	60.74
767924143	TKT	TKT	Transketolase isoform X1	0.94	**0.59**	**0.68**	8	12	60.69
295986608	IGLL5	IGLL5	Immunoglobulin lambda-like polypeptide 5 isoform 1	1.10	**1.97**	1.64	8	29	60.47
74271845	A2ML1	A2ML1	Alpha-2-macroglobulin-like protein 1 isoform 1 precursor	0.29	0.54	**1.99**	2	57	60.33
189458821	TGM3	TGM3	Protein-glutamine gamma-glutamyltransferase E	0.67	**0.39**	**0.54**	8	30	60.18
768031399	ARSA	ARSA	Arylsulfatase A isoform X1	0.70	**0.18**	0.46	2	1	59.63
145279222	C251	WDR66	WD repeat-containing protein 66 isoform 1	**6.47**	4.25	0.73	2	1	59.13
301172750	MUC5B	MUC5B	Mucin-5B precursor	2.39	**1.97**	0.96	8	145	59.04
42794752	ACSL3	ACSL3	Long-chain-fatty-acid--CoA ligase 3	**2.81**	0.17	0.10	2	1	58.55
4503107	CYTC	CST3	Cystatin-C precursor	1.45	**2.06**	1.55	8	15	58.06
4506925	SH3L1	SH3BGRL	SH3 domain-binding glutamic acid-rich-like protein	1.26	**0.27**	0.47	2	3	57.54
768038036	SH3L1	SH3BGRL	SH3 domain-binding glutamic acid-rich-like protein isoform X2	0.41	0.89	**3.14**	4	2	57.54
114199475	VPS41	VPS41	Vacuolar protein sorting-associated protein 41 homolog isoform 1	1.59	**1.82**	0.90	6	1	57.30
120433590	ACBP	DBI	Acyl-CoA-binding protein isoform 3	1.27	0.58	**0.52**	2	2	57.22
767950128	DEF3	DEFA3	Neutrophil defensin 3 isoform X1	0.78	**0.46**	0.57	8	4	57.20
767910129	S10A8	S100A8	Protein S100-A8 isoform c	**2.24**	**0.38**	**0.19**	2	13	56.94
300244562	CRIS3	CRISP3	Cysteine-rich secretory protein 3 isoform 2 precursor	1.20	**1.48**	1.33	8	9	56.57
158937236	PSA	NPEPPS	Puromycin-sensitive aminopeptidase	**0.23**	0.92	4.40	2	11	56.15
153251272	CPPED	CPPED1	Serine/threonine-protein phosphatase CPPED1 isoform b	0.43	**0.25**	0.58	2	1	56.14
40549418	PERL	LPO	Lactoperoxidase isoform 1 preproprotein	1.11	**1.61**	**1.42**	8	41	55.98
331999954	K2C4	KRT4	Keratin, type II cytoskeletal 4	0.37	**0.37**	0.96	8	20	55.60
767903808	AMY1B	AMY1B	Alpha-amylase 1B precursor	**2.86**	**3.20**	1.13	8	932	55.60
507588248	ACTN2	ACTN2	Alpha-actinin-2 isoform 2	0.81	**7.97**	5.96	2	2	55.50
45827734	SPR1A	SPRR1A	Cornifin-A	0.84	1.24	**1.69**	6	4	55.32
4502085	AMYP	AMY2A	Pancreatic alpha-amylase precursor	1.94	**2.29**	1.12	6	654	55.08
4502337	ZA2G	AZGP1	Zinc-alpha-2-glycoprotein precursor	1.40	**2.14**	1.56	8	46	54.01
50659080	AACT	SERPINA3	Alpha-1-antichymotrypsin precursor	1.19	**1.76**	1.20	8	5	52.28
71361688	PRTN3	PRTN3	Myeloblastin precursor	**0.40**	**0.33**	0.80	4	11	51.42
767939339	ILEU	SERPINB1	Leukocyte elastase inhibitor isoform X1	0.71	**0.65**	1.13	6	17	51.19
767939343	ILEU	SERPINB1	Leukocyte elastase inhibitor isoform X2	**0.60**	0.84	1.33	2	17	51.19
7706635	CRNN	CRNN	Cornulin	**0.55**	**0.38**	0.76	8	20	50.71
4504529	HIS1	HTN1	Histatin-1 precursor	**0.07**	0.90	**3.21**	4	11	50.08
4507725	TTHY	TTR	Transthyretin precursor	0.43	1.31	**3.02**	2	3	49.84
40068518	6PG	PGD	6-phosphogluconate dehydrogenase, decarboxylating isoform 1	0.50	**0.44**	0.93	2	24	49.00
751130505	6PGD	PGD	6-phosphogluconate dehydrogenase, decarboxylating isoform 2	**0.45**	**0.31**	0.69	2	22	49.00
4507261	STAT	STATH	Statherin isoform a precursor	2.18	**7.20**	0.98	8	119	36.72

Color coding and intensity illustrates the variations in protein ratios (green < 1, yellow close to 1, red > 1). The bold numbers indicate the protein is significant at *p* < 0.05. Ratios: B/N: benign/normal; M/N: malignant/normal; M/B: malignant/benign; runs: number of samples in 8-plex; AUC: area under the curve.

**Table 2 ijms-24-04164-t002:** Significant serum proteins identified in normal, benign and malignant samples.

Accession #	UniProt ID	Gene ID	Protein Name	B/N	M/N	M/B	Runs	Peptides (95%)	AUC (%)
4502443	DYST	DST	Dystonin isoform 1e precursor	**3.65**	1.93	0.52	2	2	95.80
105990532	APOB	APOB	Apolipoprotein B-100 precursor	**0.93**	**0.93**	1.04	8	456	95.42
4504349	HBB	HBB	Hemoglobin subunit beta	0.78	**1.97**	**2.04**	6	5	93.73
4557871	TRFE	TF	Serotransferrin precursor	**0.25**	**0.19**	0.77	2	38	93.59
55743122	RET4	RBP4	Retinol-binding protein 4 precursor	1.40	**1.48**	1.03	8	54	93.48
936697130	PROS	PROS1	Vitamin K-dependent protein S isoform 1 precursor	**0.78**	1.04	1.28	8	20	93.48
578822814	A2MG	A2M	Alpha-2-macroglobulin isoform X1	**0.96**	**0.94**	0.97	8	684	92.39
767989245	CADH5	CDH5	Cadherin-5 isoform X1	**1.50**	1.18	0.80	8	5	91.60
89191868	VWF	VWF	Von Willebrand factor preproprotein	**1.57**	0.90	**0.59**	8	17	91.58
73858570	IC1	SERPING1	Plasma protease C1 inhibitor precursor	0.90	**0.76**	0.89	8	54	88.63
530366456	C4BPA	C4BPA	C4b-binding protein alpha chain isoform X1	**0.66**	0.87	1.29	8	16	86.41
7669550	VINC	VCL	Vinculin isoform meta-VCL	0.61	1.75	**2.86**	2	2	85.53
45580688	CO7	C7	Complement component C7 precursor	0.98	0.72	**0.68**	8	61	85.14
4503143	CATD	CTSD	Cathepsin D preproprotein	**0.35**	0.72	2.03	2	1	82.09
767934633	CO6	C6	Complement component C6 isoform X4	**1.60**	1.32	**0.79**	8	58	81.67
115298678	CO3	C3	Complement C3 preproprotein	0.97	**0.95**	0.96	8	809	80.78
32189392	PRDX2	PRDX2	Peroxiredoxin-2	**1.89**	**2.16**	1.11	6	9	80.24
530416417	APOC1	APOC1	Apolipoprotein C-I isoform a precursor	0.59	0.28	**0.36**	8	3	76.02
32130518	APOC2	APOC2	Apolipoprotein C-II precursor	0.69	**0.69**	1.06	8	23	75.30
15811782	GPKOW	GPKOW	G-patch domain and KOW motifs-containing protein	1.1	0.19	**0.17**	2	2	74.64
223671861	PROP	CFP	Properdin precursor	**1.40**	1.33	0.90	8	11	73.84
38016947	CO5	C5	Complement C5 isoform 1 preproprotein	**0.68**	0.75	1.08	6	73	72.95
40548420	COL11	COLEC11	Collectin-11 isoform b	**0.31**	0.55	1.73	2	2	72.86
73858566	HEP2	SERPIND1	Heparin cofactor 2 precursor	**0.64**	**0.54**	0.78	8	30	71.61
167857790	A1AG1	ORM1	Alpha-1-acid glycoprotein 1 precursor	2.39	**3.47**	1.31	8	88	69.22
21071030	A1BG	A1BG	Alpha-1B-glycoprotein precursor	**1.66**	**1.54**	0.93	8	183	67.10
514239916	CO8B	C8B	Complement component C8 beta chain isoform 1 preproprotein	0.58	**0.54**	1.00	6	26	66.92
4557321	APOA1	APOA1	Apolipoprotein A-I isoform 1 preproprotein	0.96	**0.91**	**0.94**	8	202	66.64
530375762	CPN2	CPN2	Carboxypeptidase N subunit 2 isoform X1	0.88	**0.74**	0.83	8	18	66.55
16418467	A2GL	LRG1	Leucine-rich alpha-2-glycoprotein precursor	1.26	**1.49**	1.17	8	26	66.10
4501987	AFAM	AFM	Afamin precursor	**1.43**	0.97	0.67	8	67	63.81
4502419	BLVRB	BLVRB	Flavin reductase (NADPH)	0.94	**3.13**	3.31	2	1	62.79
11321561	HEMO	HPX	Hemopexin precursor	**0.92**	0.97	0.99	8	347	62.75
73858564	CBG	SERPINA6	Corticosteroid-binding globulin precursor	0.71	**0.60**	0.83	8	13	62.28
4505047	LUM	LUM	Lumican precursor	**1.82**	1.40	0.79	8	25	61.56
4503689	FIBA	FGA	Fibrinogen alpha chain isoform alpha-E preproprotein	**1.37**	**2.03**	1.34	8	23	61.18
4504893	KNG1	KNG1	Kininogen-1 isoform 2 precursor	1.21	0.83	**0.7**	8	102	59.04
8923909	ERBIN	ERBIN	Erbin isoform 2	2.26	0.78	**0.34**	2	1	58.76
33350932	DYHC1	DYNC1H1	Cytoplasmic dynein 1 heavy chain 1	**0.22**	0.37	1.64	2	2	57.99
573459745	KAIN	SERPINA4	Kallistatin isoform 2 precursor	**0.67**	**0.64**	0.91	8	20	57.95
530365618	CD5L	CD5L	CD5 antigen-like isoform X1	1.32	1.65	**1.73**	6	4	56.55
60499001	SOX	PIPOX	Peroxisomal sarcosine oxidase	0.35	**0.23**	0.66	2	1	55.97
767975372	CE290	CEP290	Centrosomal protein of 290 kDa isoform X6	**2.83**	2.81	0.98	2	3	55.70
40317626	TSP1	THBS1	Thrombospondin-1 precursor	**0.56**	0.76	1.30	6	30	55.48
767985152	TSP1	THBS1	Thrombospondin-1 isoform X2	**0.55**	**0.30**	**0.58**	2	29	55.48
767953771	MED30	MED30	Mediator of RNA polymerase II transcription subunit 30 isoform 3	0.52	**0.28**	0.82	6	4	54.62
4502337	ZA2G	AZGP1	Zinc-alpha-2-glycoprotein precursor	1.04	**1.36**	1.22	8	67	54.01
156523970	FETUA	AHSG	Alpha-2-HS-glycoprotein preproprotein	1.71	**1.34**	0.86	8	149	52.63
205277441	THBG	SERPINA7	Thyroxine-binding globulin precursor	**0.66**	**0.60**	0.80	8	20	52.62
189163530	A1AT	SERPINA1	Alpha-1-antitrypsin precursor	0.12	**0.08**	0.77	2	1	52.04
4502149	APOA2	APOA2	Apolipoprotein A-II preproprotein	0.90	**0.50**	**0.50**	8	119	51.83
190194360	PCD20	PCDH20	Protocadherin-20 precursor	0.62	2.74	**4.41**	2	1	51.39
530374534	HRG	HRG	Histidine-rich glycoprotein isoform X1	**0.77**	1.04	1.36	2	49	50.77
4507725	TTHY	TTR	Transthyretin precursor	**1.47**	1.31	0.97	8	134	49.84
4885399	SMC3	SMC3	Structural maintenance of chromosomes protein 3	0.69	**0.31**	0.45	2	1	49.08
153266841	APOH	APOH	Beta-2-glycoprotein 1 precursor	**1.58**	**1.75**	1.09	8	124	45.16

Color coding and intensity illustrates the variations in protein ratios (green < 1, yellow close to 1, red > 1). The bold numbers indicate that the protein is significant at *p* < 0.05. Ratios: B/N: benign/normal; M/N: malignant/normal; M/B: malignant/benign; runs: number of samples in 8-plex; AUC: area under the curve.

**Table 3 ijms-24-04164-t003:** Characteristics of subjects used for iTRAQ analysis. Saliva and blood samples were obtained prior to any treatments received by the subjects.

Subject ID	Age	Diagnosis	Subject ID	Age	Diagnosis	Subject ID	Age	Diagnosis
N1	41	NAD	B1	53	Fibroadenoma	M1	75	Invasive ductal carcinoma, Grade II, ER+/PR+/HER2-
N2	54	NAD	B2	46	Ductal ectasia	M2	64	Invasive ductal carcinoma, Grade I, ER+/PR+/HER2-
N3	50	NAD	B3	59	Ductal hyperplasia; microcalcifications associated with benign ducts	M3	43	Invasive ductal carcinoma, Grade II, ER+/PR+/HER2-
N4	47	NAD	B4	52	Benign fibroepithelial lesion	M4	71	Invasive ductal carcinoma, Grade II, ER+/PR+/HER2-
N5	63	NAD	B5	38	Ductal hyperplasia	M5	68	Invasive ductal carcinoma, Grade III, ER+/PR-/HER2-
N6	52	NAD	B6	44	Stromal fibrosis	M6	51	Invasive ductal carcinoma, Grade III, ER-/PR-/HER2-
N7	48	NAD	B7	64	Atypical lobular dysplasia	M7	49	Invasive ductal carcinoma, Grade II, ER+/PR+/HER2-
N8	65	NAD	B8	52	Ductal hyperplasia	M8	49	Invasive ductal carcinoma, Grade II, ER+/PR+/HER2-
Mean ± SD	52.5 ± 8.1		Mean ± SD	51 ± 8.3		Mean ± SD	58.8 ± 12.1	
Range	41–65		Range	38–64		Range	43–75	

NAD: no abnormality detected on mammogram; N1–N8: normal subjects; B1–B8: subjects with benign breast disease; M1–M8: subjects with malignant breast disease.

## Data Availability

Any data and R script in this study can be obtained from the corresponding author upon reasonable request. In this study, publicly available datasets were also analyzed for ROC curve and prognostic performance analyses. These are available as the BRCA project on The Cancer Genome Atlas (https://portal.gdc.cancer.gov/ accessed on 30 December 2022).

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
