# Peer review of "Potential Early Markers for Breast Cancer: A Proteomic Approach Comparing Saliva and Serum Samples in a Pilot Study"

_ijms, 2023, doi:10.3390/ijms24044164_

Round 1

Reviewer 1 Report

The authors present a pilot study using iTrac platform of proteomic analysis comparing normal, benign and malignant samples. 

The potential strength of the study is related to the technology used. Unfortunately the reason for using iTrac vs other proteomic strategies which have been previously published, and there are many, need to be clearly outlined. A 2022 meta-analysis of proteomic analyses of saliva for breast cancer detection is a good start (Koopaie et al, Cancer Med 2022). 

Weaknesses

Very small sample size, indicating that any results are at best hypothesis generating

Providing p values with 6 normals, 13 benign and 15 malignant seems inappropriate

Lack of demographics: what does benign mean? what were the ages and race/ethnicities? Ductal vs lobular cancer? Stage, grade, treatment before sample collection? W/o this information, the results are potentially meaningless

Seems the figures related to clinical relevance are all in the supplemental section. None of this matters unless the findings are clinically meaningful, so suggest add the figures to the main paper

Author Response

We thank the reviewer for the critically review of our manuscript. We have responded to the comments and made the changes accordingly in the manuscript using track changes.

Reviewer 2 Report

The authors sought to identify protein markers in saliva and serum for early detection of breast cancer. A thorough analysis of individual saliva and serum samples from women without breast disease and women diagnosed with benign or malignant breast disease was performed using isobaric labels for the method of relative and absolute quantitation (iTRAQ) and random effects model. A total of 591 and 371 proteins, respectively, were identified in saliva and serum samples from the same individuals. The authors' systematic approach illustrates a feasible platform for investigating a responsive proteomic profile in benign and malignant breast diseases using saliva and serum from the same women.

1. I did not see information on how the presence of non-malignant and malignant pathologies of the mammary glands was confirmed.

2. There is no information about the type of breast cancer, histology, molecular biological characteristics, stage. This is important for understanding whether the sample was homogeneous.

3. Did the age in the compared groups differ? the indicated range of 25-85 years is very wide for such a small sample.

Author Response

We thank the reviewer for critical review of our manuscript. We have responded to the comments in a point-by-point manner and have made the changes in the manuscript using track changes.

Round 2

Reviewer 1 Report

I do not understand the demographics table. Microcalcifications is not a pathologic diagnosis. I request pathologic diagnoses please. For individuals with breast cancer, I request stage of disease, as well as ER/PR/HER 2 status.

Author Response

I do not understand the demographics table. Microcalcifications is not a pathologic diagnosis. I request pathologic diagnoses please. For individuals with breast cancer, I request stage of disease, as well as ER/PR/HER 2 status.

Response: We thank the reviewer for the comment. We have carefully examined the surgical pathology reports from Pathology Department again on the patients with benign and malignant diagnoses. The final diagnosis, grade and ER/PR/HER2 status were duly noted and have been updated in Table 3 on pages 8/9 in the revised manuscript. This has certainly improved the characteristics of the subjects and we sincerely hope the reviewer will find the updated version satisfactory.

Reviewer 2 Report

I have no more remarks/comments on the article. I believe that in its present form the manuscript can be recommended for publication.

Author Response

I have no more remarks/comments on the article. I believe that in its present form the manuscript can be recommended for publication.

Response: We are thankful to the reviewer for their recommendation.